# Overcoming Intrinsic and Acquired Cetuximab Resistance in RAS Wild-Type Colorectal Cancer: An In Vitro Study on the Expression of HER Receptors and the Potential of Afatinib

**DOI:** 10.3390/cancers11010098

**Published:** 2019-01-15

**Authors:** Ines De Pauw, Filip Lardon, Jolien Van den Bossche, Hasan Baysal, Patrick Pauwels, Marc Peeters, Jan Baptist Vermorken, An Wouters

**Affiliations:** 1Center for Oncological Research (CORE), University of Antwerp, 2610 Wilrijk, Belgium; filip.lardon@uantwerpen.be (F.L.); jolien.vandenbossche@uantwerpen.be (J.V.d.B.); hasan.baysal@uantwerpen.be (H.B.); patrick.pauwels@uza.be (P.P.); marc.peeters@uza.be (M.P.); janb.vermorken@uza.be (J.B.V.); an.wouters@uantwerpen.be (A.W.); 2Department of Pathology, Antwerp University Hospital, 2650 Edegem, Belgium; 3Department of Oncology, Antwerp University Hospital, 2650 Edegem, Belgium

**Keywords:** afatinib, cetuximab resistance, colorectal cancer, EGFR, HER receptors

## Abstract

The epidermal growth factor receptor (EGFR) is an important therapeutic target in colorectal cancer (CRC). After the initial promising results of EGFR-targeted therapies such as cetuximab, therapeutic resistance poses a challenging problem and limits the success of effective anti-EGFR cancer therapies in the clinic. In order to overcome resistance to these EGFR-targeted therapies, new treatment options are necessary. The objective of this study was to investigate the expression of human epidermal growth factor (HER) receptors and the efficacy of afatinib, a second-generation irreversible EGFR-tyrosine kinase inhibitor, in *RAS* wild-type CRC cell lines with different cetuximab sensitivities. CRC cell lines with different sensitivities to cetuximab showed rather low EGFR expression but high HER2 and HER3 expression. These results were in line with the The Cancer Genome Atlas (TCGA) data from CRC patients, where higher mRNA levels of HER2 and HER3 were also detected compared to EGFR. Therefore, the targets of afatinib were indeed expressed on the CRC cell lines used in this study and in CRC patients. Furthermore, cetuximab resistance had no significant influence on the expression levels of HER receptors in CRC cell lines (*p* ≥ 0.652). This study also demonstrated that afatinib was able to induce a concentration-dependent cytotoxic effect in *RAS* wild-type CRC cell lines with different cetuximab sensitivities. Neither cetuximab resistance (*p* = 0.233) nor hypoxia (*p* = 0.157) significantly influenced afatinib’s cytotoxic effect. In conclusion, our preclinical data support the hypothesis that treatment with afatinib might be a promising novel therapeutic strategy for CRC patients experiencing intrinsic and acquired cetuximab resistance.

## 1. Introduction

Increased or sustained signaling of the epidermal growth factor receptor (EGFR) plays an integral role in the tumorigenesis of many cancer types, including colorectal cancer (CRC), making it a compelling drug target. Inhibition of EGFR signaling has been a focus of research over the last decades and this has led to the development of multiple EGFR-targeted agents. The EGFR-targeted monoclonal antibodies (mAbs) cetuximab and panitumumab have already demonstrated a significant survival improvement in patients with *RAS* wild-type metastatic CRC (mCRC) when given in combination with FOLFIRI (leucovorin, 5-fluorouracil (5-FU) and irinotecan) and with FOLFOX (leucovorin, 5-FU, and oxaliplatin), respectively [1,2,3,4,5,6,7]. Initially, these therapies were given to unselected populations, but novel insights indicated that both cetuximab and panitumumab are only effective in wild-type *RAS* patients [8]. In *RAS* wild-type mCRC, the addition of cetuximab to FOLFIRI and panitumumab to FOLFOX resulted in a median overall survival of 23.5 months and 25.8 months versus 19.5 months and 20.2 months with chemotherapy alone, respectively [9,10]. Nevertheless, even in *RAS* wild-type disease, 40–60% of patients fail to respond, possibly due to mechanisms that can compensate for reduced EGFR signaling or mechanisms that can modulate EGFR-dependent signaling [1,11,12,13,14,15,16,17]. Therefore, new therapeutic strategies are necessary in order to improve treatment outcomes of mCRC patients. 

The precise mechanisms of intrinsic and acquired resistance to EGFR inhibitors remain unclear. Since EGFR signaling is prominent in CRC, the inhibition of this EGFR pathway is still considered as an important therapeutic strategy. Extensive dimerization among the different human epidermal growth factor (HER) receptor tyrosine kinases suggests that blocking signaling from more than one family member may be essential to effectively treat CRC and limit drug resistance [18]. In contrast to the first-generation EGFR inhibitors, afatinib is an irreversible tyrosine kinase inhibitor that blocks EGFR as well as HER2 and HER4 [19,20,21]. As HER3 requires heterodimerization with other HER-family receptors, afatinib inhibits HER3 as well. This leads to an increased inhibition of HER-receptor signaling and a more complete blockade of EGFR signaling [22]. Consequently, treatment with afatinib holds the potential to result in a distinct and more pronounced therapeutic benefit. 

Our previous preclinical research showed not only that afatinib displays a cytotoxic effect in CRC, but also demonstrates effective cytotoxic activity in intrinsic and acquired cetuximab-resistant head and neck squamous cell carcinoma (HNSCC) cell lines [23,24]. However, we alluded already to the possibility of cross-resistance between cetuximab and afatinib. Therefore, in this study, we planned to investigate the potential of afatinib to overcome cetuximab resistance in CRC and the possibility of cross-resistance. Despite these optimistic preclinical results, afatinib treatment has not yet led to a major clinical benefit in CRC patients. Hence, identification of predictive biomarkers is key to further explore the efficacy of afatinib in selected CRC patients.

This study aims to provide preclinical data concerning the expression of HER receptors and the potential of afatinib in a panel of *RAS* wild-type CRC cell lines that are either sensitive or have intrinsic/acquired resistance to cetuximab. With this in mind, we decided to: (1) examine the expression of HER receptors in CRC in order to determine the presence of afatinib’s targets, (2) test the influence of cetuximab resistance on the expression of HER receptors in *RAS* wild-type CRC cell lines, (3) determine the cytotoxic effect of afatinib in these *RAS* wild-type CRC cell lines with different cetuximab sensitivities, (4) study the efficacy of afatinib under both normal and reduced oxygen conditions, as CRC is often characterized by regions with reduced oxygen levels and as there is a link between hypoxia and EGFR signaling [25], (5) examine the molecular mechanisms underlying the cytotoxic effect of afatinib, and (6) explore the potential synergistic interactions between afatinib and irinotecan.

## 2. Results

### 2.1. Identification of Intrinsically Cetuximab-Resistant CRC Cell Lines and Generation of Acquired Cetuximab-Resistant Cell Lines

Sensitivity to cetuximab therapy was investigated in a panel of *RAS* wild-type CRC cell lines (Figure 1A). Based on the dose–response curves and the corresponding half maximal inhibitory concentration (IC_50_) values, two out of four CRC cell lines (i.e., SW48 and HT29) were considered as intrinsically resistant to cetuximab, as the percentage of viable cells in these cell lines did not decrease below 50%. CaCo2 and Lim1215 were identified as cetuximab-sensitive, as the IC_50_ values (17.69 ± 7.59 nM and 0.12 ± 0.04 nM, respectively) are considered as clinically relevant [26]. Next, cell lines with acquired cetuximab resistance (i.e., CaCo2-R and Lim1215-R) were generated from initially cetuximab-sensitive cell lines. As shown in Figure 1B, cetuximab treatment did not lead to a dose-dependent effect in CaCo2-R and Lim1215-R, in contrast to the cetuximab-sensitive CaCo2-PBS and Lim1215-PBS cells. The stability of cetuximab resistance was confirmed after culture in drug-free medium for 6 weeks (Figure 1C).

### 2.2. CRC Cell Lines and Patients Show Higher Expression of HER2 and HER3 than EGFR

As afatinib binds to multiple members of the HER receptor family, the basal cellular membrane protein expression level of these HER family members was determined in our panel of CRC cell lines with different sensitivities to cetuximab. 

EGFR expression was observed in all the CRC cell lines used in this study. However, the percentage of EGFR-positive cells ranged between 11% and 34% (Figure 2A). Moreover, the intensity of EGFR expression on these EGFR-positive cells was rather low (Δ mean fluorescence intensities (MFI) ranging between 88 ± 20 and 444 ± 94) (Figure 2B). In contrast to EGFR, all cell lines demonstrated high percentages of HER2 and HER3-positive cells (68–96% and 57–88%, respectively) and the expression level of both HER2 and HER3 on receptor-positive cells was also high (ΔMFI ranging between 1659 ± 333 and 6412 ± 491 and ΔMFI ranging between 1290 ± 160 and 4537 ± 1490, respectively) (Figure 2). No significant differences in percentages of EGFR, HER2, and HER3-positive cells (*p* ≥ 0.652) and ΔMFI of receptor-positive cells (*p* ≥ 0.695) were observed between cetuximab-sensitive, intrinsically-resistant, and acquired cetuximab-resistant CRC cell lines. HER4 expression was barely observed in any of the CRC cell lines tested and when detected, HER4 expression levels were very low (data not shown).

Next, RNA sequencing data from the The Cancer Genome Atlas (TCGA) dataset of CRC patients (Provisional, RNASeqV2 RSEM, 382 sequenced patients) was used to compare our in vitro results [27,28]. The mRNA expression levels of EGFR, HER2, and HER3 in tumor samples of CRC patients are shown in Figure 3. CRC patients showed significantly higher HER2 and HER3 mRNA levels compared to EGFR (*p* < 0.001 and *p* < 0.001, respectively). This corresponds with our flow cytometric findings in CRC cell lines. Moreover, CRC patients also demonstrated very low HER4 mRNA expression, which is in line with our flow cytometric data in CRC cell lines (data not shown). 

Overall, these results demonstrate that CRC cell lines with different sensitivities to cetuximab show rather low EGFR expression but high HER2 and HER3 expression. These results were in line with the TCGA data from CRC patients, as also in patient material, whereby higher mRNA levels of HER2 and HER3 were detected compared to EGFR. Furthermore, cetuximab resistance had no influence on the expression levels of HER receptors in CRC cell lines. We therefore concluded that the targets of afatinib were indeed expressed on the CRC cell lines used in this study and in CRC patients.

### 2.3. Intrinsic and Acquired Cetuximab-Resistant CRC Cell Lines Are Sensitive to Afatinib Treatment

The cytotoxic effect of the irreversible HER family blocker afatinib was studied in cetuximab-sensitive, intrinsically/acquired-resistant, and PBS-treated control CRC cell lines under both normoxic and hypoxic conditions. A clear concentration-dependent cytotoxic effect of afatinib (0–10 μM) after 72 hours of treatment was observed in all CRC cell lines (Figure 4). The IC_50_ values for afatinib under normoxia ranged from 0.007 ± 0.002 μM to 2.379 ± 0.869 μM (Table 1). Intrinsically and acquired cetuximab-resistant CRC cell lines tended to show higher IC_50_ values compared to cetuximab-sensitive CRC cell lines Lim1215 and CaCo2. This suggested the possibility of cross-resistance between cetuximab and afatinib. However, no significant effect of cetuximab resistance on afatinib’s cytotoxicity was observed (*p* = 0.233). Furthermore, no statistical difference in the cytotoxic effect of afatinib was observed when cells were treated under hypoxic conditions (*p* = 0.157).

Overall, afatinib showed a clear concentration-dependent cytotoxic effect in cetuximab-sensitive and intrinsically-resistant CRC cell lines and CRC cell lines with acquired resistance. Our results demonstrated that neither cetuximab resistance nor exposure to hypoxia provoked therapeutic resistance to afatinib.

### 2.4. Treatment of CRC Cell Lines with Afatinib Has Little Influence on the Cell Cycle Distribution and the Induction of Apoptotic Cell Death

We studied the effect of afatinib on the distribution of CRC cells in the different phases of the cell cycle was investigated (Figure 5). SW48 and Lim1215-PBS cells demonstrated a small yet significant increase in the percentage of G_0_/G_1_ cells (*p* = 0.049 and *p* = 0.037, respectively) and a decrease in the percentage of S phase cells (*p* = 0.007 and *p* = 0.068, respectively). A significant decrease in the percentage of S cells was also found in CaCo2-R (*p* ≤ 0.033). In contrast, a significant increase in the percentage of G_2_/M cells was observed in CaCo2 (*p* ≤ 0.028), CaCo2-PBS (*p* = 0.039) and Lim1215-R (*p* = 0.047) after treatment with high dose of afatinib (IC_60_ and IC_80_). Overall, afatinib did not have a pronounced effect on the cell cycle distribution of CRC cells, independently of their sensitivity to cetuximab. 

Furthermore, the induction of apoptotic cell death after afatinib treatment was studied using Annexin V-fluorescein isothiocyanate (AnnV-FITC)/propidium iodide (PI) staining (Figure 6). This technique identifies cells that are AnnV+/PI− or AnnV+/PI+, which is an indication for apoptotic cell death. SW48 cells demonstrated a significant increase in AnnV+/PI− and AnnV+/PI+ cells as well as a corresponding decrease of viable AnnV−/PI− cells after 72 hours of treatment with afatinib (Figure 6A). However, in the majority of CRC cell lines used in this study, no significant induction of apoptotic cell death was observed after treatment with afatinib concentrations up to IC_80_ (Figure 6).

### 2.5. The Combination Treatment of Afatinib with Irinotecan Leads to Additive Effects in CRC Cell Lines

In order to investigate the potential interaction between afatinib and irinotecan, CRC cells were incubated with fixed doses of afatinib for 72 hours, followed or preceded by treatment with 0–50 μM irinotecan for 24 h. The fixed afatinib concentrations were based on the outcome of the monotherapy experiments and corresponded to the IC_20_ and IC_40_ values specific for each cell line. The dose–response curves after treatment with both sequential combination regimens are shown in Figure 7 and Figure 8. All CRC cell lines were sensitive to treatment with irinotecan monotherapy, with IC_50_ values ranging from 0.09 ± 0.02 μM to 40.07 ± 4.59 μM, not including CaCo2-PBS and CaCo2-R (Table 2). Concerning CaCo2-PBS and CaCo2-R, IC_50_ values could not be calculated as irinotecan established a concentration-dependent cytotoxic effect, yet the percentage of viable cells did not decrease below 50%. Compared to irinotecan treatment as monotherapy, treatment with afatinib before irinotecan demonstrated no significant decrease in IC_50_ value (*p* ≥ 0.051), except in the HT29 cell line (*p* = 0.023). Furthermore, the combination index (CI) ranged from 0.81 ± 0.10 to 1.12 ± 0.11, indicating an additive interaction. Similarly, treatment with irinotecan followed by afatinib showed no significant decrease in IC_50_ value (*p* ≥ 0.116) and CI ranged from 0.72 ± 0.13 to 1.22 ± 0.32, revealing additive to subadditive effects. Thus, sequential exposure to afatinib followed by irinotecan or the reverse regimen (i.e., irinotecan followed by afatinib) revealed additive, yet no synergistic interactions.

## 3. Discussion

Targeted therapies are the key for the personalized treatment of cancer patients. After initial promising results of EGFR-targeted therapies such as cetuximab, therapeutic resistance poses a challenging problem and limits the success of effective anti-EGFR cancer therapies in the clinic. As a result, new treatment options are needed to overcome drug resistance. Due to the intensive interactions between HER receptors, inhibition of one HER receptor can be compensated by other HER family members, which therefore must be targeted by new therapeutic regimens. In contrast to the first-generation EGFR-inhibitors, afatinib irreversibly blocks EGFR, HER2, and HER4. Consequently, we hypothesized that treatment with afatinib might result in a more pronounced therapeutic benefit, even in patients who experience resistance to first-generation EGFR inhibitors. To investigate this hypothesis, we first examined the expression of HER receptors in CRC in order to determine the presence of afatinib’s targets. In addition, the influence of cetuximab resistance on the expression of HER receptors in *RAS* wild-type CRC cell lines was determined. Afterwards, the cytotoxicity of afatinib, as a single agent and in combination with irinotecan, was examined in this panel of *RAS* wild-type CRC cell lines that are either sensitive or have intrinsic/acquired resistance to cetuximab.

In the first part of this study, we determined the expression of HER receptors in *RAS* wild-type CRC cell lines with different cetuximab sensitivities and compared these results with RNA sequencing data from the TCGA dataset of CRC patients. According to the target expression of EGFR and HER2, the CRC cell lines used in this study are valid candidates for treatment with afatinib. These in vitro results were in line with the TCGA data from CRC patients, who also express EGFR and HER2, indicating the presence of afatinib’s targets in CRC. Interestingly, both CRC cell lines and patients demonstrated higher HER2 and HER3 expression compared to EGFR expression. It has already been suggested that HER2 expression might play an important role in response to afatinib treatment in CRC [29,30]. Conflicting results have been reported concerning the correlation of HER2 and HER3 expression with clinicopathological characteristics and prognosis of CRC patients. Whereas several studies have demonstrated that expression of HER2 and HER3 is correlated with clinicopathological factors and poor prognosis [31,32], results from the CALGB 80,203 trial showed that high tumor mRNA levels of HER2 is a prognostic marker associated with longer progression free survival across all mCRC patients in this study [33]. In addition, other studies have even reported that there is no correlation between HER2 and HER3 expression and clinicopathological characteristics as well as prognostic factors [34,35,36,37]. Therefore, it remains an open question whether HER2 and HER3 expression can be used as a prognostic marker in CRC.

Resistance to cetuximab has been associated with extensive dimerization among the different HER receptor tyrosine kinases [18]. In the present study, it was shown that cetuximab resistance had no significant influence on the expression of HER receptors in CRC cell lines. Nevertheless, the kinase activity of these receptors could still be strongly induced, provoking resistance to cetuximab [18]. Therefore, evaluation of the influence of cetuximab resistance on the phosphorylation levels of HER receptors would also be highly interesting to study more in depth. Even more relevant would be the extensive characterization of the intrinsically and acquired cetuximab-resistant CRC cell lines in order to verify the expression of different markers and signaling pathways, such as MET activation [38], which will provide essential information for the identification of cetuximab resistance mechanisms in CRC.

Several studies have demonstrated a correlation between HER2 amplification and acquired resistance to cetuximab in CRC [39,40]. Furthermore, studies such as the CALGB 80,203 trial demonstrated that high HER3 levels are associated with both resistance and lack of benefit from cetuximab in mCRC [33,41]. However, Seligmann et al. has recently reported that HER3 expression in *KRAS* wild-type CRC patients was predictive for efficacy of panitumumab [42]. Although these data are conflicting, they show the potential role of HER expression in guiding the response of tumors to anti-EGFR treatments. At the moment, however, such evidence about the predictive value of HER receptor expression does not allow definitive conclusions, and additional prospective studies are needed [43]. Nevertheless, inhibition of homo-and heterodimerization of these HER receptors remains a promising strategy to overcome cetuximab resistance.

In the second part of this study, we demonstrated that afatinib has the potential to overcome intrinsic and acquired cetuximab resistance, as it was able to establish a cytotoxic effect in CRC cell lines with different sensitivities to cetuximab. Camidge et al. have recently demonstrated in a phase Ib trial that the maximum plasma concentration of afatinib at its maximum tolerated dose was 313 ng/mL, which corresponds to 0.644 μM [44]. In our study, only the IC_50_ values of SW48, HT29, and CaCo2-R cell lines are higher than this maximum afatinib plasma concentration. In the future, it would be highly interesting to further investigate the efficacy of afatinib in long-term in vitro assays. For example, in the past, Herr et al. have already determined the long-term effect of afatinib treatment on the HT29 CRC cell line in a clonogenic assay (1 μM, 14 days) [45]. They reported that afatinib significantly inhibited long-term cell growth of the HT29 cells, indicating that afatinib also has a long-term effect on cell growth. In a next step, implementation of in vivo models, patient-derived xenografts and organoids would be warranted, as there are several limitations to in vitro cell line studies that limit the direct translation of this work to the clinical arena. Nevertheless, we are convinced that such laboratory studies may contribute to the optimization of treatment protocols in the clinic in at least two different ways. First, in vitro studies can provide an important platform for selecting potentially promising drugs. Second, in vitro studies may help to improve the time schedule of novel combination regimens. As such, though extrapolation of in vitro data to the clinic should be considered with caution, we believe that our experiments can provide a strong experimental basis for further development in an in vivo and clinical setting.

Concerning the molecular mechanisms underlying afatinib’s cytotoxic effect in CRC, previous studies demonstrated that afatinib caused a G_0_/G_1_ cell cycle arrest and induced apoptotic cell death in different cancer cell lines, including CRC cell lines [23,30,46,47,48]. In the present study, however, treatment with afatinib did not lead to very pronounced G_0_/G_1_ cell cycle arrest or induction of apoptosis. Therefore, analyzing caspase activity and markers of autophagy and other types of cell death after afatinib treatment, would be highly interesting to study more in depth. Thus, additional research is necessary to further unravel the mode of action of afatinib in CRC. Importantly, exposure to hypoxia did not provoke therapeutic resistance to afatinib in CRC cell lines. This is an interesting finding as oxygen deficiency is a common characteristic of CRC and these hypoxic tumor regions often contain viable cells that are more resistant to conventional therapies [49].

Despite the fact that statistical analysis did not reveal a significant effect of cetuximab resistance on the cytotoxicity of afatinib, we noticed that intrinsically and acquired cetuximab-resistant CRC cell lines tended to show higher IC_50_ values compared to cetuximab-sensitive CRC cell lines, thus suggesting the possibility of cross-resistance between cetuximab and afatinib. Previous studies in HNSCC have indicated that afatinib is more effective in patients whose tumors are cetuximab-naïve [23,50,51]. Consequently, resistance to EGFR inhibitors is not exclusively due to alterations of HER receptor signaling; other signaling pathways could also play a pivotal role. It is important to take this into account in future studies.

In the past few years, afatinib has been investigated in various clinical trials. In phase I studies, some promising signs of antitumor activity after afatinib treatment were observed in patients with advanced solid tumors, including CRC. After these promising phase I dose-escalation studies, afatinib was further investigated in several phase II studies. Hickish et al. reported that afatinib (40 mg/day) demonstrated inferior response and survival compared to cetuximab in *KRAS* wild-type CRC patients [52]. Despite the promising preclinical data on the inhibitory effect of afatinib in specific *KRAS*-mutated CRC cell lines, no clinical benefit of afatinib was observed in *KRAS* mutated patients in this phase II study. Previous treatment history might have had an impact on the lack of clinical benefit observed in the study. Recently, interest in afatinib as monotherapy for the treatment of CRC is diminished, indicated by the low number of novel clinical trials in CRC patients found on ClinicalTrials.gov. However, afatinib did demonstrate promising clinical efficacy in other solid tumors, including HNSCC and non-small-cell lung carcinoma (NSCLC) [51,53,54].

As most adjuvant cancer treatments are combinations of chemotherapeutic agents and/or radiotherapy, we are convinced that new EGFR-targeted agents, such as afatinib, will achieve their greatest efficacy in combination with traditional cytotoxic agents and/or radiotherapy. As cetuximab has been approved for the treatment of mCRC in combination with the irinotecan containing chemo-regimen FOLFIRI, we investigated the combination of afatinib with irinotecan in CRC cell lines that are sensitive and have intrinsic/acquired resistance to cetuximab. In clinical studies, it has already been reported that simultaneous treatment with afatinib and standard chemotherapy is associated with increased frequency of side effects [55,56]. Therefore, we found it more relevant to study sequential treatment regimens. In our study, sequential exposure to afatinib followed by irinotecan or the inverse regimen (i.e., irinotecan followed by afatinib) revealed additive, yet not synergistic interactions. Therefore, it would be highly interesting to further investigate the effect of afatinib with other conventional CRC chemotherapeutics, such as oxaliplatin and 5-FU. When molecular mechanism of afatinib’s cytotoxic effect are better understood, rationally designed combination strategies can be further developed.

Besides combining afatinib with standard chemotherapy and/or radiotherapy, combinations with other targeted agents such as vascular endothelial growth factor inhibitors have been investigated in clinical studies, with limited success in CRC patients [57,58]. In addition, the combination of two EGFR-targeted therapies, i.e. afatinib and cetuximab, has been examined in patients with advanced solid tumors. This study recently reported that adverse events were manageable and anti-tumor activity was observed in some patients, particularly in those with NSCLC and HNSCC [59]. Furthermore, a preclinical study showed that BRAF inhibition leads to upregulation of a variety of receptor tyrosine kinases in CRC cell lines, including EGFR, HER2 and HER3 [45]. Combination of BRAF inhibitors with inhibitors dually targeting EGFR and HER2 such as afatinib significantly reduced metabolic activity and proliferative potential of CRC cells [45]. These encouraging results prove that investigating molecular mechanisms indeed leads to the development of rational combination strategies that can be further examined in clinical studies. As such, regimens combining EGFR-targeted therapies with other targeted therapies are complex but of particular interest. For a comprehensive review on preclinical and clinical studies on afatinib in monotherapy and in combination regimens in CRC, we refer to De Pauw et al. [24]. 

As mentioned above, interest in afatinib for the treatment of CRC has diminished, and currently no novel studies have been initiated to investigate combinations of afatinib with other drugs for the treatment of CRC patients (ClinicalTrials.gov). However, as our and other preclinical studies have demonstrated the potential of afatinib in CRC cell lines, its limited clinical benefit in CRC patients could be explained by the lack of predictive biomarkers necessary for optimal patient selection. We foresee that liquid biopsies can play a key role in the progress needed for improving the efficacy of targeted therapies in CRC. Liquid biopsies will help to identify and monitor the biomarkers for both response and resistance to EGFR-targeted therapies in CRC as well as speed up the process of patient selection [43]. As such, the identification of predictive biomarkers is essential to identify CRC patients that would benefit most from afatinib treatment. After that, afatinib might be picked up again for further novel clinical studies in CRC patients.

## 4. Materials and Methods 

### 4.1. Cell Culture

The human CRC cell lines SW48, HT29, and CaCo2 were purchased from the American Type Culture Collection (ATCC, Rockville, MD, USA). Lim1215 was obtained from CellBank Australia (Westmead NSW, Australia). All cell lines were *RAS* wild-type. SW48, HT29, and CaCo2 were cultured in DMEM medium, while Lim1215 was grown in RPMI medium (Life Technologies, Merelbeke, Belgium). Each medium was supplemented with 10% fetal bovine serum (FBS), 1% penicillin/streptomycin and 1% L-glutamine (Life Technologies, Merelbeke, Belgium). Sodium pyruvate was additionally added to RPMI medium. Cells were grown as monolayers and maintained in exponential growth in 5% CO_2_/95% air in a humidified incubator at 37 °C. Authenticity of cell lines was verified by establishing short tandem repeat profile. All cell lines were confirmed free of mycoplasma infection through regular testing (MycoAlert Mycoplasma Detection Kit, Lonza, Verviers, Belgium).

### 4.2. Cytotoxicity Assays

Cell survival was determined using the colorimetric sulforhodamine B (SRB) assay, as previously described [60,61]. This endpoint assay assesses the number of viable cells after treatment, as it is not possible to make a distinction with this assay between inhibition of proliferation (cytostatic effect) and cell death (cytotoxic effect). Optimal seeding densities for each cell line were determined in order to ensure exponential growth during the whole duration of the assay. Cells were counted automatically with a Scepter 2.0 device (Merck Millipore SA/NV, Overijse, Belgium). After overnight incubation at 37 °C, cells were treated with cetuximab alone (0–100 nM, 168 h), afatinib alone (0–10 μM, 72 h), or afatinib in combination with irinotecan (0–50 μM, 24 h). Hereby, two sequential combination schedules were tested: Afatinib for 72 h immediately followed by irinotecan for 24 h;Irinotecan for 24 h immediately followed by afatinib for 72 h.

The pharmaceuticals, cetuximab (anti-EGFR mAb, Merck, Darmstadt, Germany) and irinotecan (Selleck Chemicals, Houston, TX, USA) were diluted in sterile PBS. Afatinib (EGFR-tyrosine kinase inhibitor, Selleck Chemicals) was diluted in DMSO (Merck Millipore SA/NV, Overijse, Belgium) and further dilutions were made in cell culture medium. Survival rates were calculated by: (mean optical density (OD) of treated cells/mean OD of control cells) × 100%. IC_50_ values (i.e., drug concentration causing 50% growth inhibition) were calculated using WinNonlin software (Pharsight, Mountain View, CA, USA). Possible synergism between afatinib and irinotecan was determined by calculation of the combination index (CI) using the ‘Additive Model’ as described by others [62,63,64]. CI < 0.8, CI = 1.0 ± 0.2 and CI > 1.2 indicate synergism, additivity, and antagonism, respectively.

### 4.3. Oxygen Conditions

Hypoxia (1% O_2_) was achieved in a Bactron IV anaerobic chamber (Shel Lab, Cornelius, OR, USA), as described previously [65]. After overnight incubation to allow attachment of cells, hypoxic conditions were initiated immediately after addition of the drug. Measurements with ToxiRae II air oxymeter (Rae Benelux, Hoogstraten, Belgium) confirmed that the oxygen tension in the gas phase was stable at 1% O_2_.

### 4.4. Generation of Resistant Cell Clones

Generation of cell clones with acquired resistance was performed as described previously [23,66,67]. Acquired cetuximab-resistant variants were derived from the original cetuximab-sensitive parental CaCo2 and Lim1215 cell lines by continuous exposure to cetuximab, starting with the IC_50_ concentration of cetuximab for 7 days. In parallel, control parental cells were exposed to the vehicle control (PBS). After 10 dose doublings, dose–response studies were determined for each resistant cell line (suffix R). In order to examine whether acquired resistance was a transient or permanent effect, dose-response curves of cetuximab were re-assessed in the resistant cell lines after 6 weeks in culture without cetuximab.

### 4.5. Expression Analysis of HER Family Members

The baseline extracellular protein expression level of EGFR, HER2, HER3, and HER4 was assessed using flow cytometry, as previously described [23,67]. Cells were incubated with EGFR, HER2, HER3, and HER4 phycoerythrin (PE)-conjugated antibodies (10 μL/10^6^ cells, 25μg antibody in 1ml, R&D Systems, Minneapolis, MN, USA). Corresponding isotype controls (respectively: rat IgG2A, mouse IgG2B, mouse IgG1, and mouse IgG2A, 10 μL/10^6^ cells, 50 μg antibody in 2 mL, R&D Systems) were included for all samples and served as negative controls. Dead cells were excluded from the analysis by staining with 7-AAD (BD Biosciences, Erembodegem, Belgium). All samples were measured on a FACScan flow cytometer (BD Biosciences). Flow cytometric data were analyzed using FlowJo v10.1 (TreeStar Inc., Ashland, OR, USA). The percentages of EGFR, HER2, HER3, and HER4-positive cells (Overton) were determined in comparison with the corresponding isotype control using the Overton method. Furthermore, the signal for aspecific binding was subtracted from the MFI (=ΔMFI). This parameter indicates the amount of extracellular expression of EGFR, HER2, HER3, and HER4 on individual cells. 

The obtained results were compared using the RNA sequencing data from the TCGA dataset of CRC patients (Provisional, 382 sequenced patients). RNASeqV2 from TCGA was processed and normalized using the software package RSEM (RNA-Seq by Expectation Maximization) to generate transcripts per million. This dataset was downloaded from cBioportal.

### 4.6. Assays for Apoptosis and Cell Cycle Distribution

After overnight incubation, cells were treated for 72 h with afatinib. Since the sensitivity to afatinib strongly varied between the cell lines, afatinib concentrations were based on the outcome of the monotherapy experiments and corresponded with the IC_20_, IC_40_, IC_60,_ and IC_80_ values specific for each cell line. Cell cycle distribution was determined using the CycleTEST^TM^ PLUS DNA reagent kit (BD Biosciences). Induction of apoptotic cell death was investigated flow cytometrically using the AnnV-FITC/PI assay (BD Biosciences). Both assays were performed on a FACScan flow cytometer and analyzed with FlowJo v10.1. 

### 4.7. Statistical Analysis

All experiments were performed at least three times independently, unless otherwise stated. In cytotoxicity experiments, each condition was tested in triplicate in each of the three independent experiments. Flow cytometry experiments were independently performed three times with one sample for each condition. Results are presented as mean  ± standard deviation. The effects of various conditions and treatments were studied using linear regression or linear mixed models in case of non-independent observations. All models were fitted using a stepwise backward strategy, starting from a model with all fixed effects and their interactions. If the interaction term was not significant, a model with only the main effects was fitted. If the treatment effect was significant, a Dunnet posthoc analysis was performed. 

The effect of cetuximab resistance on the expression of HER receptors was analyzed using a linear mixed model with cetuximab resistance status as fixed effect. A random intercept for cell line was added, in order to account for the dependence between observations within the same cell line. Significant differences in HER receptor mRNA expression of CRC patients (RNASeqV2 TCGA data) were assessed using one-way ANOVA with Bonferroni posthoc analysis.

Effects of oxygen and resistance status on afatinib’s cytotoxic effect were modeled using a linear mixed model with oxygen status, resistance status, and their interaction as fixed effects. A random intercept for cell line was added to account for the dependence between observations within the same cell line. Significant differences in the cell cycle distribution, induction of apoptotic cell death and IC_50_ values in the combination experiments were investigated using one-way ANOVA with Dunnet posthoc analysis.

GraphPad Prism 7 was used for data comparison and artwork. All statistical analyses were performed in JMP Pro 13 and SPSS v24. *p*-values below 0.050 were considered significant.

## 5. Conclusions

This study demonstrated that *RAS* wild-type CRC cell lines with different sensitivities to cetuximab show rather low EGFR expression but high HER2 and HER3 expression. These results were in line with the TCGA data from CRC patients, where higher mRNA levels of HER2 and HER3 were also detected compared to EGFR. Consequently, the targets of afatinib are expressed on the CRC cell lines used in this study and in CRC patients. Cetuximab resistance had no influence on the expression levels of HER receptors in CRC cell lines. The present study also demonstrated that afatinib was able to establish a concentration-dependent cytotoxic effect in *RAS* wild-type CRC cell lines that are either sensitive or have intrinsic/acquired resistance to cetuximab. Neither cetuximab resistance nor hypoxia significantly influenced afatinib’s cytotoxic effect. Treatment with afatinib had little effect on the cell cycle distribution and the induction of apoptosis. Sequential combinations of afatinib with irinotecan demonstrated additive effects, yet no synergistic interactions. Although these preclinical data support the hypothesis that afatinib might be a promising novel therapeutic strategy for the treatment of *RAS* wild-type CRC patients experiencing cetuximab resistance, to date no survival benefit has been observed in clinical trials. Therefore, additional studies with biomarker-driven patient recruitment are required to further explore the efficacy of afatinib in CRC patients.

## Figures and Tables

**Figure 1 cancers-11-00098-f001:**
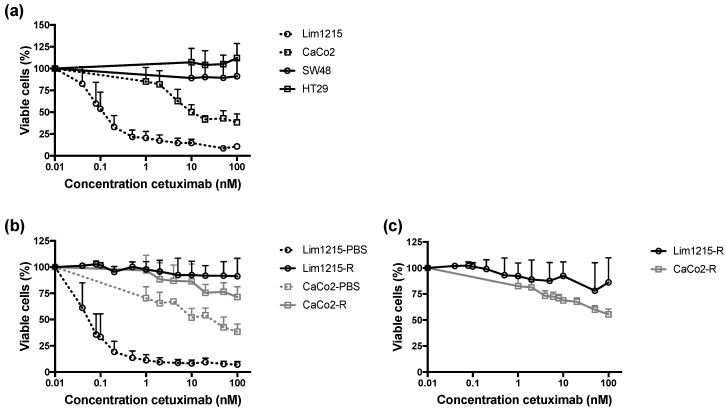
Sensitivity to cetuximab treatment. (**a**) Dose–response curves for *RAS* wild-type colorectal cancer (CRC) cell lines after cetuximab treatment (168 h). Two out of four CRC cell lines (i.e., SW48 and HT29) were considered as intrinsically resistant to cetuximab. Cetuximab-sensitive CRC cell lines (i.e., Lim1215 and CaCo2) were used to generate isogenic cell lines with acquired cetuximab resistance. (**b**) Dose–response curves for isogenic CRC cell lines with acquired cetuximab resistance (Lim1215-R and CaCo2-R) and cetuximab-sensitive (Lim1215-PBS and CaCo2-PBS) CRC cell lines. (**c**) Dose–response curves for the CRC cell lines with acquired cetuximab resistance after 6 weeks of culture in drug-free medium, followed by cetuximab treatment for 168 h. This graph represents two independent experiments executed in threefold.

**Figure 2 cancers-11-00098-f002:**
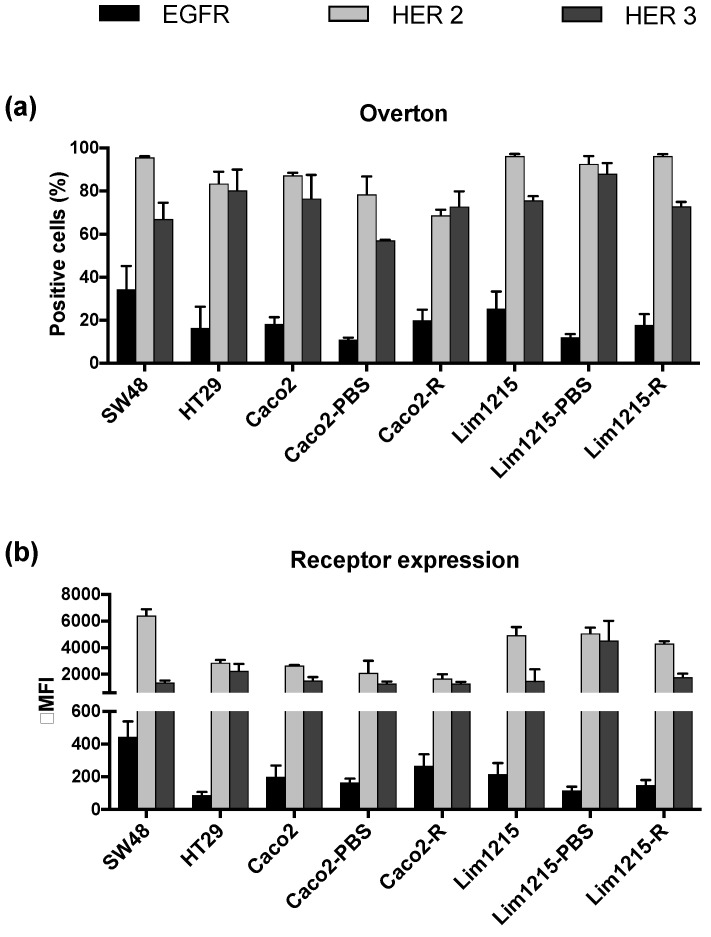
Human epidermal growth factor (HER) receptor expression in CRC cell lines with different sensitivities to cetuximab. (**a**) The percentage of EGFR, HER2, and HER3-positive cells (overton), determined using FACScan flow cytometer. (**b**) The expression levels of EGFR, HER2, and HER3 on the corresponding receptor-positive cells (Δ mean fluorescence intensities (MFI)), determined using FACScan flow cytometer. EGFR: epidermal growth factor receptor.

**Figure 3 cancers-11-00098-f003:**
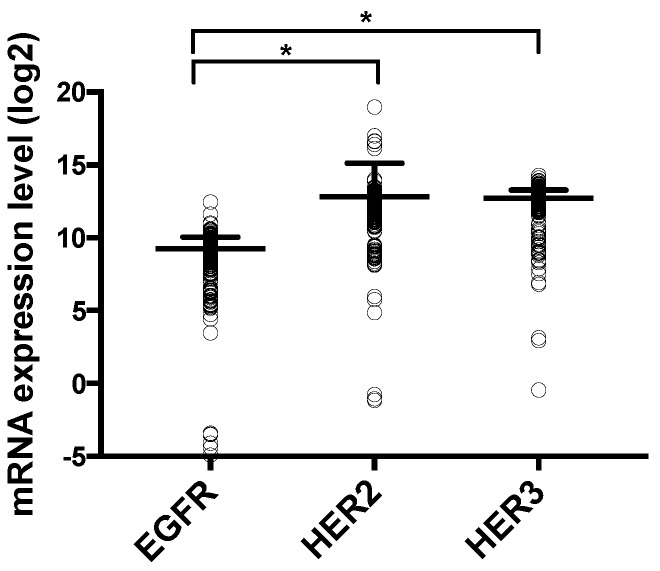
mRNA expression level of HER receptors in CRC patients, available from The Cancer Genome Atlas (TCGA). The graph shows the mRNA expression (mean and standard deviation) of EGFR, HER2, and HER3 from 382 CRC patients (individual dots). This dataset (TCGA Provisional, RNASeqV2) was downloaded from cBioportal. * *p*-value ≤ 0.050.

**Figure 4 cancers-11-00098-f004:**
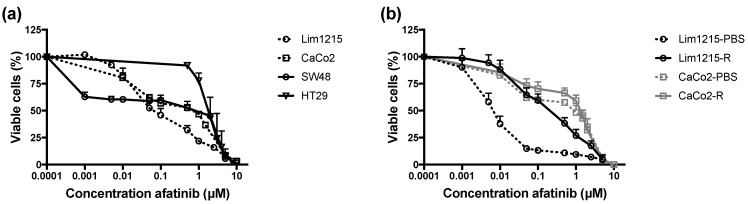
Cytotoxicity of afatinib in CRC cell lines with different cetuximab sensitivities. (**a**) Dose–response curves for cetuximab-sensitive and intrinsically-resistant CRC cell lines. (**b**) Dose–response curves for acquired cetuximab-resistant and corresponding isogenic cetuximab-sensitive CRC cell lines.

**Figure 5 cancers-11-00098-f005:**
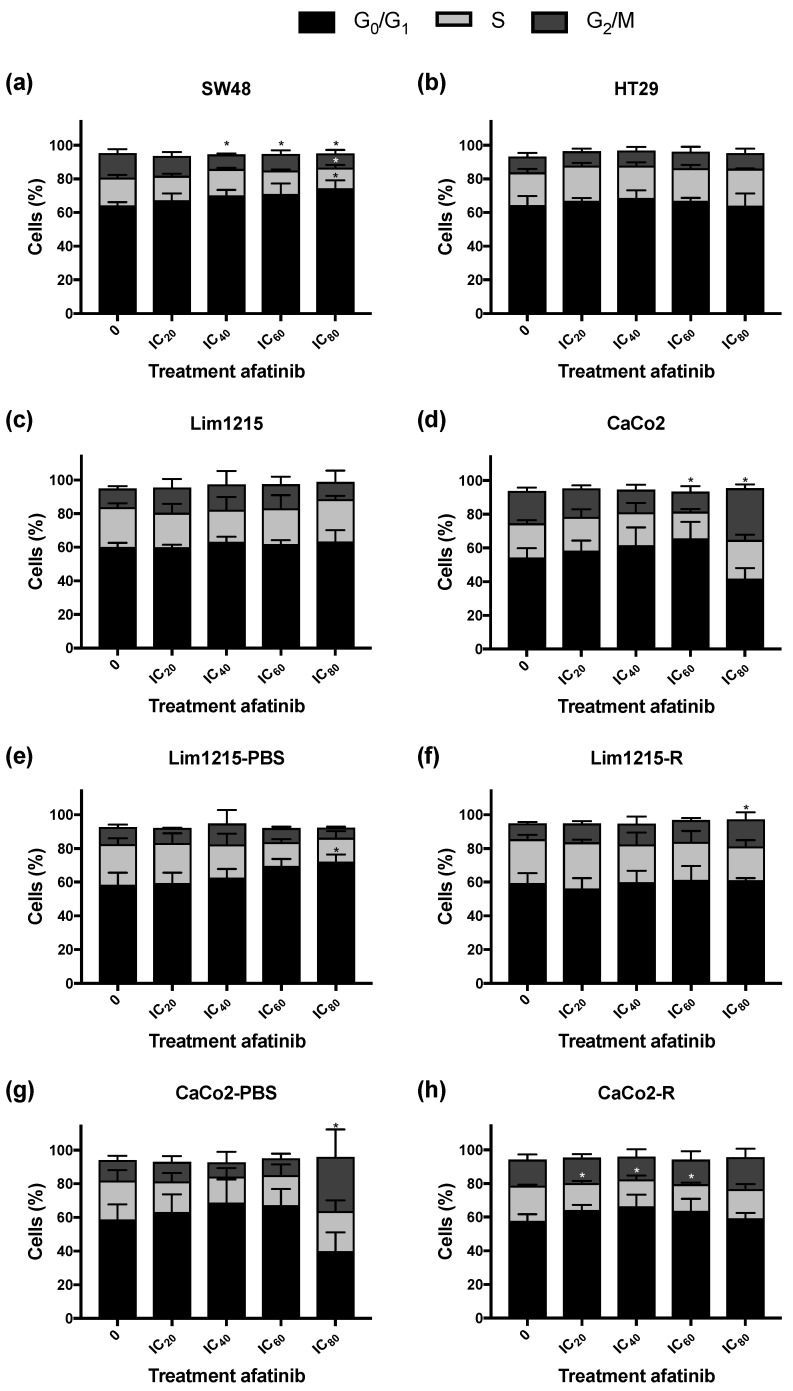
Cell cycle distribution in CRC cell lines after afatinib treatment (72 h, 0–IC_80_). DNA content was measured using flow cytometry after staining the cells with propidium iodide (PI). Cells were divided into three groups: G_0_/G_1_ phase (2n), S phase (2n–4n) and G_2_/M phase (4n). The effect of afatinib treatment on the cell cycle distribution of intrinsically cetuximab-resistant (**a**,**b**), cetuximab-sensitive (**c**,**d**), acquired cetuximab-resistant, and corresponding isogenic cetuximab-sensitive (**e**–**h**) CRC cell lines. * *p*-value treatment effect on the percentage of cells ≤0.050.

**Figure 6 cancers-11-00098-f006:**
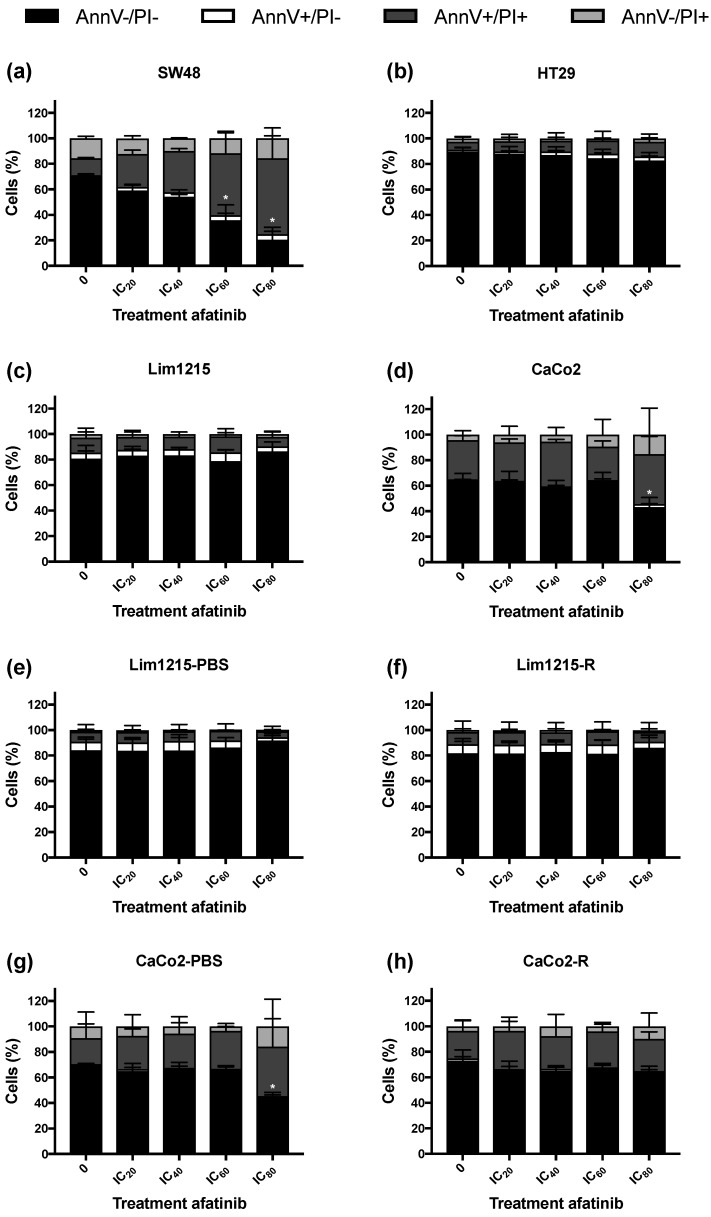
Induction of apoptotic cell death in CRC cell lines after afatinib treatment (72 h, 0–IC_80_). Cells were stained with Annexin V-fluorescein isothiocyanate (AnnV-FITC) and PI and measured using flow cytometry. The effect of afatinib treatment on the induction of apoptotic cell death of intrinsically cetuximab-resistant (**a**,**b**) cetuximab-sensitive (**c**,**d**), acquired cetuximab-resistant, and corresponding isogenic cetuximab-sensitive (**e**–**h**) CRC cell lines. * *p*-value treatment effect on the percentage of cells ≤0.050.

**Figure 7 cancers-11-00098-f007:**
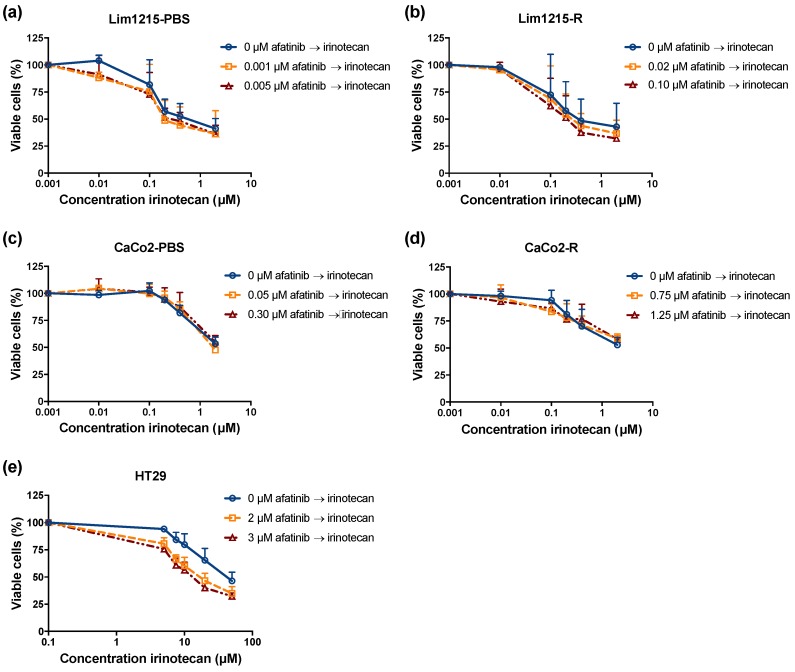
The cytotoxic effect of afatinib followed by irinotecan in CRC cell lines with different sensitivities to cetuximab. Dose–response curves for the cetuximab-sensitive (**a**,**c**), corresponding isogenic acquired cetuximab-resistant (**b**,**d**), and intrinsically cetuximab-resistant (**e**) CRC cell lines indicate an additive effect. Survival curves are corrected for the cytotoxic effect of 72 h of afatinib alone. Cells were treated with fixed concentrations afatinib, which were based on the outcome of the monotherapy experiments.

**Figure 8 cancers-11-00098-f008:**
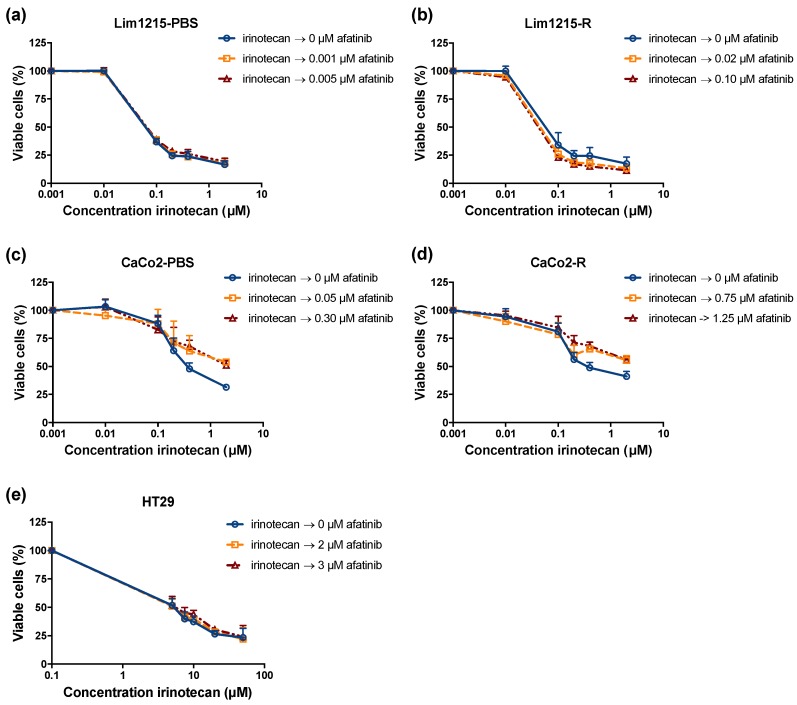
The cytotoxic effect of irinotecan followed by afatinib in CRC cell lines with different sensitivities to cetuximab. Dose–response curves for the cetuximab-sensitive (**a**,**c**), corresponding isogenic acquired cetuximab-resistant (**b**,**d**) and intrinsically cetuximab-resistant (**e**) CRC cell lines indicate an additive to subadditive effect. Survival curves are corrected for the cytotoxic effect of 72 h afatinib alone. Cells were treated with fixed concentrations afatinib, which were based on the outcome of the monotherapy experiments.

**Table 1 cancers-11-00098-t001:** Half maximal inhibitory concentration (IC_50_) values and standard errors for CRC cell lines after incubation with afatinib for 72 h under normoxic and hypoxic conditions.

IC_50_ Afatinib 72 h (μM)			
Cell Line	Cetuximab Resistance Status	Normoxia (21% O_2_)	Hypoxia (1% O_2_)
Lim1215	Sensitive	0.081 ± 0.021	0.178 ± 0.055
CaCo2	Sensitive	0.341 ± 0.199	0.604 ± 0.307
SW48	Intrinsically resistant	2.379 ± 0.869	2.109 ± 0.691
HT29	Intrinsically resistant	1.805 ± 0.041	1.816 ± 0.117
Lim1215-PBS	PBS-treated control, sensitive	0.007 ± 0.002	0.010 ± 0.003
Lim1215-R	Acquired resistance	0.174 ± 0.030	0.458 ± 0.060
CaCo2-PBS	PBS-treated control, sensitive	0.591 ± 0.384	0.260 ± 0.197
CaCo2-R	Acquired resistance	1.570 ± 0.264	1.398 ± 0.270

**Table 2 cancers-11-00098-t002:** IC_50_, combination index (CI), and standard errors for CRC cell lines after sequential treatment with afatinib followed by irinotecan as well as irinotecan followed by afatinib. CI < 0.80, CI = 1.00 ± 0.20 and CI > 1.20 indicated synergism, additivity, or antagonism, respectively.

Cell Line	Condition	IC_50_ (μM)	*p*-Value	CI
Lim1215-PBS	72 h 0 μM afatinib→24 h irinotecan	0.28 ± 0.26	/	/
	72 h 0.001 μM afatinib→24 h irinotecan	0.36 ± 0.19	0.757	0.92 ± 0.12
	72 h 0.005 μM afatinib→24 h irinotecan	0.39 ± 0.16	0.681	0.94 ± 0.08
	24 h irinotecan→72 h 0μM afatinib	0.11 ± 0.04	/	/
	24 h irinotecan→72 h 0.001 μM afatinib	0.09 ± 0.04	0.994	1.02 ± 0.04
	24 h irinotecan→72 h 0.005 μM afatinib	0.10 ± 0.04	0.488	1.10 ± 0.07
Lim1215-R	72 h 0 μM afatinib→24 h irinotecan	1.48 ± 0.99	/	/
	72 h 0.02 μM afatinib→24 h irinotecan	0.64 ± 0.36	0.458	0.90 ± 0.07
	72 h 0.1 μM afatinib→24 h irinotecan	0.44 ± 0.19	0.417	0.81 ± 0.10
	24 h irinotecan→72 h 0 μM afatinib	0.09 ± 0.02	/	/
	24 h irinotecan→72 h 0.02 μM afatinib	0.08 ± 0.02	0.235	0.79 ± 0.10
	24 h irinotecan→72 h 0.1 μM afatinib	0.06 ± 0.02	0.116	0.72 ± 0.13
CaCo2-PBS	72 h 0 μM afatinib→24 h irinotecan	ND	/	/
	72 h 0.05 μM afatinib→24 h irinotecan	ND	/	1.00 ± 0.07
	72 h 0.3 μM afatinib→24 h irinotecan	ND	/	1.03 ± 0.03
	24 h irinotecan→72 h 0 μM afatinib	0.46 ± 0.15	/	/
	24 h irinotecan→72 h 0.05 μM afatinib	ND	/	1.22 ± 0.32
	24 h irinotecan→72 h 0.3 μM afatinib	ND	/	1.22 ± 0.29
CaCo2-R	72 h 0 μM afatinib→24 h irinotecan	ND	/	/
	72 h 0.75 μM afatinib→24 h irinotecan	ND	/	0.99 ± 0.08
	72 h 1.25 μM afatinib→24 h irinotecan	ND	/	1.00 ± 0.09
	24 h irinotecan→72 h 0 μM afatinib	0.55 ± 0.28	/	/
	24 h irinotecan→72 h 0.75 μM afatinib	ND	/	1.14 ± 0.20
	24 h irinotecan→72 h 1.25 μM afatinib	ND	/	1.22 ± 0.18
HT29	72 h 0 μM afatinib→24 h irinotecan	40.07 ± 4.59	/	/
	72 h 2 μM afatinib→24 h irinotecan	18.96 ± 3.29	0.051	1.12 ± 0.11
	72 h 3 μM afatinib→24 h irinotecan	14.73 ± 2.77	0.023	1.10 ± 0.07
	24 h irinotecan→72 h 0 μM afatinib	4.34 ± 1.06	/	/
	24 h irinotecan→72 h 2 μM afatinib	4.74 ± 0.61	0.993	1.02 ± 0.06
	24 h irinotecan→72 h 3 μM afatinib	5.52 ± 0.69	0.566	1.10 ± 0.07

ND: Not determined as cell survival did not decrease below 50%. *p* ≤ 0.050, significant difference in IC_50_ compared to irinotecan monotherapy./: cannot be calculated.

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
