# Peer review of "Overcoming Intrinsic and Acquired Cetuximab Resistance in RAS Wild-Type Colorectal Cancer: An In Vitro Study on the Expression of HER Receptors and the Potential of Afatinib"

_cancers, 2019, doi:10.3390/cancers11010098_

Round 1

Reviewer 1 Report

In the present manuscript De Pauw et al investigate the expression of HER receptors on a panel of RAS wild type CRC cell lines and the potential efficacy of afatinib treatment regardless of cetuximab sensitivity. Although this work does not bring so much novelty neither from the methodological / experimental point of view nor for the scientific results, it might be of a potential interest for the scientific community.

Please find below some comments and questions that need to be addressed before considering this manuscript for publication.

1.     CaCo2 cells have been previously indicated as cetuximab-resistant cells because overexpression of the MET or PDGFRA receptors (NA Song et al, Int J Mol Sci. 2014; Medico et al, Nat Comm 2015). Could you please evaluate the expression of these receptor in your CACO2 cells and better clarify their role in terms of sensitivity/resistance to cetuximab? Which IC50 do the authors consider clinically relevant?

2.     A more detailed characterization of the prepared resistant cell lines Lim1215-R and CaCo2-R is missing. Desirable could be analysis of expression of different markers and molecular signaling analysis.

3.     The statistics in Fig.1b is missing.

4.     Resistant CaCo2-R cells are approximately 3-fold more resistant, and Lim1215-R are almost 25-fold more resistant to afatinib in normoxia conditions compared to their parental counterparts (-PBS) as stated in the Table 1. It is also obvious from Fig.4b that at least in Lim1215-R cells there is clearly very different concentration-dependent cytotoxic effect of afatinib compared to parental cells. Why do you consider this as a potential cross-resistance, and how is it possible that the statistics was not significant? Moreover, the curve for LIM1215 in Fig4a and LIM1215-PBS in Fig2b should look quite similar while they evidently differ. Which explanation could the author give?

5.     Do you have any explanation for why the results from Fig.4a and 4b do not correspond to the results from Fig.6? There is an obvious cytotoxic effect of afatinib as shown in Fig.4. Did you check different ways of programmed cell death?

6.     The authors were looking for the possible synergistic interaction of afatinib with irinotecan. Have you also tested the effect of afatinib along with other conventional CRC chemotherapeutics like oxaliplatin or 5-fluorouracil?

7.     In the conclusions section authors sum up that their preclinical data support the hypothesis that afatinib might be a promising novel therapeutic strategy for the treatment of RAS wild type CRC patients experiencing cetuximab resistance. According to me, provided data are not completely satisfactory as there is only one result (Fig.4b) showing possible afatinib efficacy. Are the toxic concentrations of afatinib reachable in human plasma? I strongly suggest to fill in another analysis proving this one result (for instance clonogenic assay, long-term efficacy of presented treatment). Adding data on either in vivo models or clinically relevant tumor models (such as patient-derived xenografts or organoids) would put much higher value this the study in order to prove the efficacy of afatinib. The authors should comment more also on the potential use of combinatorial therapies on cetuximab-resistant models.

Reviewer 2 Report

In this manuscript, the authors analyze the effect of the HER inibitor afatinib on a panel of colorectal cancer cell lines WT for KRAS, sensitive or resistant to cetuximab.

Findings presented are of interest, in the frame of the necessity to overcome clinical resistance to anti- EGFR MoAbs, which at present represents the major problem for their clinical use.

While data appear generally clean, the authors may further address few points:

·         Sensitivity to cetuximab in Figure 1 was assessed after 1 week of treatment, when cells are likely very close to confluence. Data may be shown also at 72h of treatment, in order to better confirm the model, in the light of some conflicting data on sensitivity of CACO2 cells (see for example Nature Communications volume 6, Article number: 7002, 2015.

·         Since the analyses of cell cycle and apoptotic cell death following cell treatment with afatinib fail to give conclusive data, the authors may try to analyze caspases activity and/or autophagy markers.

As a personal curiosity, I would like to have explained the label: “Overton” in Figure 2a, that I never encountered before.

Reviewer 3 Report

The manuscript demonstrated that expression levels of HER receptors were not correlate to CRC cell response to cetuximab or afatinib, suggesting that the expression levels of HER receptors cannot be used as predictive markers for the treatment of these EGFR antibody/inhibitor. Meanwhile, it validated the previous finding that afatinib works better than cetuximab, which support their hypothesis “treatment with afatinib might result in a more pronounced therapeutic benefit, even in patients who experience resistance to first-generation EGFR inhibitors. Overall, rational and experimental designs of for the manuscript are sounds. However, data presented did not provide any new information regarding EGFR-therapy and expression levels of HER receptor in CRC. Detailed comments regarding the results are:

1.       It would be more informative if the authors could determine the levels of phosphorylation of HER receptors.  

2.       The effects of cetuximab or afatinib on CRC cell viability over time (time course study) will better demonstrate CRC cell response to these drugs

3.       Fig. 8e figure legend “0 iM, 2 iM, 3iM afatinib” is confusing.

Round 2

Reviewer 1 Report

The authors have answered to many of the questions raised, although the majority of the points have been added to the discussion section and not experimentally addressed. Although I understand that some experiments such us the organoids/PDXs ones might require a significant amount of time and resources, I believe that some long-term in vitro assays such us long-term proliferation or clonogenic assays would be helpful in reinforcing the message of this work.

Sticking to point 7,  only one result (Fig.4b) shows possible afatinib efficacy in the selected cell line models, so I would recommend to add at least one of the proposed in vitro experiments. Accordingly, the discussion section (page 12, line 295) should be corrected.

Other minor comments:

-the sentence "Thus, in general, the toxic concentrations of afatinib are reachable in human plasma" should be taken out -- it sounds to generic and not properly fitting in the discussion (page 12, line 294)

-figure 8 legend: a and c are cetux-sensitive

Reviewer 3 Report

the revised manuscript addressed all comments and concerns I had. 

Round 3

Reviewer 1 Report

I'm satisfied with this last version of the manuscript.